# Insulin Resistance/Hyperinsulinemia, Neglected Risk Factor for the Development and Worsening of Heart Failure with Preserved Ejection Fraction

**DOI:** 10.3390/biomedicines12040806

**Published:** 2024-04-04

**Authors:** Serafino Fazio, Valentina Mercurio, Valeria Fazio, Antonio Ruvolo, Flora Affuso

**Affiliations:** 1Department of Internal Medicine, School of Medicine, Federico II University, Via Sergio Pansini 5, 80135 Naples, Italy; 2Department of Translational Medical Sciences, Federico II University, Via Sergio Pasini 5, 80135 Naples, Italy; valentina.mercurio@unina.it; 3UOC Medicina Interna, Azienda Ospedaliera di Caserta, 81100 Caserta, Italy; valeria.fazio@gmail.com; 4UOC Cardiologia AORN dei colli PO CTO, Viale Colli Aminei 21, 80100 Naples, Italy; ruvantonio2@gmail.com; 5Independent Researcher, Viale Raffaello 74, 80129 Naples, Italy; flora.affuso70@gmail.com

**Keywords:** insulin, insulin signaling, insulin resistance, hyperinsulinemia, cardiovascular system, cardiovascular diseases, cardiovascular risk factors, diabetes, HF, HFpEF

## Abstract

Heart failure (HF) has become a subject of continuous interest since it was declared a new pandemic in 1997 because of the exponential increase in hospitalizations for HF in the latest years. HF is the final state to which all heart diseases of different etiologies lead if not adequately treated. It is highly prevalent worldwide, with a progressive increase with age, reaching a prevalence of 10% in subjects over the age of 65 years. During the last two decades, it was possible to see that the prevalence of heart failure with preserved ejection fraction (HFpEF) was increasing while that of heart failure with reduced ejection fraction (HFrEF) was decreasing. HFpEF is typically characterized by concentric remodeling of the left ventricle (LV) with impaired diastolic function and increased filling pressures. Over the years, also the prevalence of insulin resistance (IR)/hyperinsulinemia (Hyperins) in the general adult population has progressively increased, primarily due to lifestyle changes, particularly in developed and developing countries, with a range that globally ranges between 15.5% and 46.5%. Notably, over 50% of patients with HF also have IR/Hyperins, and the percentage is even higher in those with HFpEF. In the scientific literature, it has been well highlighted that the increased circulating levels of insulin, associated with conditions of insulin resistance, are responsible for progressive cardiovascular alterations over the years that could stimulate the development and/or the worsening of HFpEF. The aim of this manuscript was to review the scientific literature that supports a pathophysiologic connection between IR/Hyperins and HFpEF to stimulate the scientific community toward the identification of hyperinsulinemia associated with insulin resistance as an independent cardiovascular risk factor in the development and worsening of HF, believing that its adequate screening in the general population and an appropriate treatment could reduce the prevalence of HFpEF and improve its progression.

## 1. Introduction

Despite the considerable progress made in the last decades in the prevention and treatment of cardiovascular diseases, they are still in first place among the causes of death. In particular, in the European community, there are approximately 2 million deaths from cardiovascular causes every year [1]. Heart failure (HF) is the culmination of cardiovascular diseases of different etiologies, and its prevalence is progressively increasing, reaching a considerable number of over 64 million people estimated to be living with it worldwide in 2017 [2,3]. Heart failure is classified based on symptoms and the ejection fraction of the left ventricle into HF with reduced ejection fraction (left ventricular ejection fraction <50%, HFrEF) and HF with preserved ejection fraction (left ventricular ejection fraction ≥50%). In recent years, the prevalence of HFpEF has progressively increased, while the prevalence of HFrEF is slowly decreasing [4]. As is well-known, HFpEF is characterized by concentric remodeling of the left ventricle and diastolic dysfunction with increased left ventricular filling pressures [4,5]. It is very likely that one of the main causes of this may be insulin resistance (IR) with an associated increase in circulating insulin levels (hyperinsulinemia), which is becoming increasingly prevalent in developed and developing countries due to evident lifestyle changes consisting mainly of an increase in caloric intake and a reduction in physical activity [6,7,8]. We believe that insulin resistance (IR)/hyperinsulinemia (Hyperins) should be considered an independent risk factor for cardiovascular disease and, in particular, an important cause of the development and progressive worsening of HFpEF. The aim of this manuscript is the review of the scientific literature that supports a pathophysiologic connection between IR/Hyperins and HFpEF to stimulate the scientific community toward the identification of hyperinsulinemia associated with insulin resistance as an independent cardiovascular risk factor in the development and worsening of HF. We believe that its adequate screening in the general population and appropriate treatment could reduce the prevalence of HFpEF and improve its progression.

For this reason, we searched the most accredited scientific databases (Pubmed, Scopus, Science Direct, etc.) for the scientific literature on the topic using the following keywords: insulin, insulin signaling, insulin resistance, hyperinsulinemia, cardiovascular system, cardiovascular diseases, cardiovascular risk factors, diabetes, HF, HFpEF.

## 2. Heart Failure

According to a recent consensus for a universal definition and classification of heart failure (HF), it was defined as a clinical syndrome characterized by symptoms and signs caused by structural and/or functional anomalies of the heart, supported by elevated levels of natriuretic peptide and/or objective evidence of systemic or pulmonary congestion [4,5,8,9]. The levels of natriuretic peptides are very variable and, generally, lower in patients with HFpEF than in patients with HFrEF, and this is easily explained by Laplace’s law. In fact, the stimulus that determines the production of these peptides in the heart, i.e., wall stress, is lower in the former than in the latter unless we are in an acute phase. Therefore, in patients with HFpEF, natriuretic peptide levels are much more variable, making it difficult to establish certain cutoff diagnostic values in such patients. Nonetheless, the i-PRESERVE and PEP-cHF studies demonstrated that both baseline NT-proBNP values and its changes from baseline have prognostic value in patients with HFpEF, improving the prediction of HF mortality and rehospitalizations [10,11]. Based on left ventricular ejection fraction (EF), HF was classified into three categories: 1. HFrEF with an EF ≤ 40%); 2. HF with mildly reduced EF (HFmrEF, with anEF between 41 and 49%; 3. HFpEF (with EF ≥ 50%). Furthermore, based on the change in EF over time, for example, following the effects of the therapy, a new category was introduced, namely HF with improved EF, which was defined as HF with a starting EF ≤ 40% that improved ≥10 points and in any case went to an EF > 40% [9]. In recent years, considering only the two main forms of HF, i.e., HFrEF and HFpEF, it was possible to notice a progressive reduction in the prevalence of the former with a progressive increase in the prevalence of the latter. In fact, in a study of patients consecutively hospitalized for HF between 1987 and 2001 at the Mayo Clinic Hospitals in Olmsted County, Minnesota, the patients with HFpEF increased from 38% to 54% [12]. Currently, HFpEF affects approximately 50% of patients with HF. HF is the main cause of hospitalization in patients aged over 65 years and accounts for between 1 and 2% of all causes of hospitalization [13]. Improving the treatment of HFpEF is becoming a priority, as it is expected to become the leading cause of HF in the coming years. HFpEF is characterized by an increase in filling pressures caused by the complex interaction of multiple components, of which the concentric remodeling of the left ventricle (LV) is a relevant component [13]. It is well known that the increase in heart stiffness resulting from aging is an important cause of diastolic dysfunction; therefore, the general aging of the population concurs with a further increase in the prevalence of HFpEF in the coming decades [14]. Furthermore, the current lack of specific therapies for this condition will make HFpEF one of the most relevant issues to be addressed. Over 50% of patients with HF also have IR with associated hyperins, and it is likely that this percentage is much higher in the setting of HFpEF [15]. In fact, it is known that diabetes is a very common comorbidity in patients with HFpEF and that, unfortunately, its presence considerably worsens the course of HF, significantly increasing the number of hospitalizations and mortality [16,17], particularly in patients with HFpEF [18,19].

## 3. Insulin Resistance/Hyperinsulinemia

Insulin resistance is a pathological condition whose prevalence has been rapidly growing in recent decades, particularly in developed and developing countries. Being slightly symptomatic or even asymptomatic, it is defined as a hidden pandemic that causes uncountable damage to the population, with relevant impact also in terms of health care systems spending [20]. It is characterized by a decreased sensitivity and metabolic response to insulin so that at given insulin levels, there are frankly reduced metabolic effects. For this reason, increased circulating insulin levels (hyperinsulinemia) are necessary to maintain blood glucose within the normal range. IR conditions are always characterized (as long as there is a sufficiently functioning pancreas) by Hyperins [7,20]. Therefore, IR/Hyperins, especially in terms of cardiovascular effects, should be considered as a single entity. It is well known that IR is a key mechanism in the pathogenesis of type 2 diabetes but also of systemic arterial hypertension, atherosclerosis, and, in particular, HFpEF, as well as other various atherosclerotic and non-atherosclerotic cardiovascular diseases [21,22,23]. The increased circulating levels of insulin (Hyperins), associated with IR conditions, adversely affect the cardiovascular system over time. Insulin has many biological targets. Beyond skeletal muscle tissue, adipose tissue and the liver, as well as the cardiovascular system [24], are relevant targets of insulin actions, as insulin receptors are highly represented in myocardial cells and vascular smooth muscle cells as well as endothelial cells [25].

The gold standard for IR diagnosis is the euglycemic/hyperinsulinemic clamp, but this test is poorly applicable for screening purposes. In fact, many indices that are easier to acquire have been created, and among these, the Homeostasis Model Assessment of IR index (HOMA-IR) and the triglyceride–glucose index (TyG) demonstrated a good correlation with the euglycemic/hyperinsulinemic clamp and could be used for screening purpose. HOMA-IR is obtained by the fasting values of blood sugar and insulin, and calculated by multiplying the fasting blood glucose value in mmol/L × the fasting insulinemia value in mU/L and dividing the result by 22.5 [(glycemia × insulinemia)/22.5]. The TyG index is calculated from fasting values of triglycerides (Tg) and blood glucose (FBG) using the following formula: [logn (serum Tg × FBG)/2]. Normal HOMA-IR values are between 0.23 and 2.5, while the cut-off value for the TyG index is 4.5 [26,27].

## 4. IR/Hyperins and HFpEF

Chronic long-lasting hyperinsulinemia, which typically characterizes IR, can damage target organs, including the cardiovascular system [24]. Insulin signaling affects the growth and survival of the heart, substrate uptake and utilization, and mitochondrial metabolism. Therefore, impaired insulin signaling may certainly contribute to the development of pathological ventricular remodeling in the heart, which can lead to the onset and/or worsening of HF [25], in particular, HFpEF. The mechanisms through which IR/Hyperins acts are multiple and not yet completely known. However, one of the best known is that insulin binds both the insulin receptor (InsR) and the insulin-like growth factor-1 receptor (IGF-1 R), and this produces the activation of signaling, which, in turn, determines, through the phosphorylation of a tyrosine kinase, the activation of two main pathways, i.e., phosphoinositide-3 kinase (PI3K/Akt), and Shc-Ras-mitogen activated protein kinase (MAPK) [28]. The PI3K/Akt pathway mainly regulates the metabolic actions of insulin, i.e., glucose metabolism in the target tissues (muscle, adipose and hepatic tissues); nitric oxide (NO) formation by vascular endothelial and smooth muscle cells is regulated by this pathway, too. The MAPKs are among the most ancient signal transduction pathways. They coordinate and regulate gene expression, mitosis, survival, apoptosis, differentiation, etc., and primarily determine the actions of stimulating cell proliferation of insulin; furthermore, they also push endothelial cells to secrete greater quantities of the vasoconstrictor endothelin-1 (ET-1) and increase the expression of adhesion molecules on the vascular endothelium [29,30,31]. In the normal subjects, these two pathways balance to obtain normal vascular homeostasis. In fact, the first pathway stimulates NO production, producing vasodilatation and reduction in vascular resistance with a consequent increase in flow to tissues. The second pathway, stimulating ET-1 formation, has vasoconstrictive actions that, together with the stimulating action on the sympathetic system, develop a hypertensive pattern and stimulate cardiac hypertrophy and atherosclerosis.

Under conditions of IR, it prevalently alters the PI3K-dependent pathway, while the MAPK-dependent pathway functioning is practically normal. For this reason, Hyperins, resulting from IR, increase the activity of the MAPK pathway, which, due to its mitogenic and proliferative actions and to the production of greater quantities of endothelin-1 (ET-1) causes chronic endothelial dysfunction with the development and progressive worsening of atherosclerosis [31,32]. Hyperins increase blood pressure not only for the increase in ET-1 levels and the stimulation of sympathetic tone but also because of the anti-natriuretic peptide’s effects of insulin [33]. In addition, the chronically increased circulating levels of insulin, for its mitogenic and proliferative effects on the cardiovascular system, stimulate the proliferation of vascular smooth muscle cells and lead to an increase in left ventricular mass (LVM) with a concentric remodeling pattern [34] (Figure 1). Such actions of Hyperins may remain misrecognized for years, at least until overt type 2 diabetes appears or a cardiovascular event occurs. One study demonstrated that IR/Hyperins is highly prevalent among nondiabetic patients with HF. In these patients, the IR indices progressively increase with the worsening of the NYHA classes of HF, and patients with HF and IR have a significantly reduced capacity for physical exercise and a peak O_2_ reached, but, unfortunately, the HF patients studied do not were distinguished in patients with HFrEF or HFpEF [16].

The scientific literature supporting the evidence that IR/Hyperins stimulate pathological remodeling of the left ventricle (LV) is extensive. Indeed, diabetic cardiomyopathy is characterized by LV hypertrophy (LVH) [35], and LVH is a very strong predictor of adverse cardiovascular events, including the development of HFpEF, as it can determine an alteration to the normal filling of the LV with diastolic dysfunction and increased filling pressures [36].

IR/Hyperins were associated with LV concentric remodeling regardless of body mass index (BMI) levels. This may lead to the hypothesis that treating IR/Hyperins could lead to a regression of LV concentric remodeling and slow the progression to HFpEF [37]. Diabetes in patients with HFpEF determines at least a double risk of hospitalizations and cardiovascular death, with a clear increase in the risk of mortality from any cause [18,19]. In fact, the results of the “Digoxin” study showed that diabetes determined a 68% increase in the risk of hospitalizations and death due to HF [38], a result subsequently confirmed also in the “Preserve Study” [39]. In the “Relax” study, it was verified that patients with HFpEF and diabetes showed significantly higher ET-1 levels and inflammation markers compared to patients with HFpEF but without diabetes [40]. As seen from this brief analysis of the literature, it is clear how the alterations caused over time at the cardiovascular level by IR/Hyperins can determine and worsen, if not corrected, HFpEF, as well as cause other adverse cardiovascular events. For this reason, we believe that it would be useful to screen the general population to promptly diagnose this pathology and treat it so as not to allow or, at least, slow down its negative evolution. Nowadays, there are both drugs and natural substances that have been shown to reduce IR/Hyperins, as well as, obviously, the possibility of changing your lifestyle for the better. Among the various substances, we will focus on only a few that we believe can produce, alone or in association, the best results: sodium-glucose cotransporter-2 inhibitors, metformin, and berberine.

## 5. Potential Treatments of IR/Hyperins

First of all, once identified, patients with IR/Hyperins should be informed of their condition, which, if not corrected, can, over time, produce important pathologies, especially affecting the cardiovascular system. They should be advised to improve their lifestyle, with a constant increase in physical activity and a reduction in caloric intake through balanced, low-carbohydrate diets [41,42]. However, in daily clinical practice, often, even the most motivated patients are unable to maintain the lifestyle change made for a long time, and, more often, they interrupt the path undertaken. For this reason, most patients with IR/Hyperins may benefit from therapeutic strategies with drugs or substances that act by improving insulin sensitivity and reducing circulating insulin levels.

A class of orally administered drugs, sodium-glucose cotransporter-2 inhibitors (SGLT2 Is), which include canaglifozin, dapaglifozin, and empaglifozin, has recently been approved and marketed for the treatment of diabetes. In addition, in Europe, sotaglifozin has been marketed, which is an inhibitor of both SGLT2 and SGLT1. It, therefore, has the additional action of reducing glucose absorption from the bowel [43]. These drugs were demonstrated in randomized, double-blind, and placebo-controlled trials to be associated with a significant reduction in the number of hospitalizations, deaths from cardiovascular events, and deaths from any cause in patients with HF [44]. For this reason, they became part of the pillars in the pharmacological treatment of HF according to the latest European guidelines [5,44]. SGLT2 Is were initially recommended by guidelines in patients with type 2 diabetes, chronic kidney disease, and eGFR ≥ 30 mL/min/1.73 m^2^ at any level of glycemic control. Guidelines were later updated on the basis of the results of seven new trials, and now, these guidelines recommend initiating therapy with SGLT2 Is in patients with an eGFR of at least 20 mL/Min/1.73 m^2^ [45].

It should be underlined that the beneficial effect of these drugs in patients with HF was also confirmed, even more strongly, in patients with HFpEF. A recent meta-analysis carried out on 12,251 patients from the “DELIVER” and “EMPEROR-Preserved” studies demonstrated that therapy with SGLT2 Is leads to a significant decrease in both deaths from cardiovascular adverse events and first-time hospitalizations for HF in patients with HFpEF [46]. Another meta-analysis of randomized controlled studies on a greater number of patients confirmed that SGLT2 Is reduces cardiovascular deaths and hospitalizations for heart failure in patients with HFpEF, similar to what happens in patients with HFrEF [47]. It has been hypothesized that the mechanisms underlying the beneficial action of these drugs in these patients are multiple (regulation of blood volume, cardiorenal and metabolic effects, actions to combat the development of pathological cardiac remodeling, favorable effects on contractility and sodium ion regulation reduction of oxidative stress and inflammation, and so on) and still not completely clarified. However, it is recognized that the loss of glucose in the urine and the reduction of circulating glucose levels activate sirtuines, which protect against oxidative stress and beneficially regulate the metabolism. Furthermore, SGLT2 Is reduce excess deposits of epicardial adipose tissue and its production of adipokines [48]. An effective treatment for HFpEF is urgently required, given the limited possibilities of therapy in this pathology. There is now sufficient clinical evidence that SGLT2 Is improve the prognosis in patients with HFpEF. However, besides such complex mechanisms, the administration of these drugs to subjects with IR/Hyperins, significantly prevalent among patients with HF, produces glycosuria and reduces the amount of circulating glucose. Consequently, the insulin levels necessary to maintain blood sugar within the normal range are drastically reduced, thus reducing the deleterious effects that Hyperins can cause in the cardiovascular system [49] [Figure 2]. Indeed, various studies demonstrated how SGLT2 Is therapy in patients with HFpEF decreases left ventricular mass (LVM) and pathological concentric remodeling, improving diastolic function and EF [50,51].

Metformin is an oral antidiabetic, commonly used for decades, belonging to the biguanide class. The scientific literature that supports the effectiveness of metformin in improving IR/Hyperins in both diabetic and non-diabetic subjects is vast [52]. A retrospective cohort analysis examined whether the use of metformin resulted in a reduction in hospitalizations and cardiovascular-related mortality using standard multivariable techniques in patients with type 2 diabetes. The conclusions of the study were that metformin therapy in such patients resulted in a reduction in morbidity and mortality related to cardiovascular risk [53].

A recent review and meta-regression analysis study clearly demonstrated that metformin treatment reduces deaths in HF patients, especially in the subgroup of patients with HfpEF (*p* < 0.003) [54]. Another meta-analysis study demonstrated that metformin reduces LVM and improves EF in both subjects with and without prior heart disease [55].

It is well known that the minute ventilation/carbon dioxide production slope (VE/VCO2 slope) is a strong predictor of prognosis among patients with HF, as the risk of mortality increases significantly if its value is >32.8 [56]. An interesting double-blind and placebo-controlled study demonstrated that metformin therapy for 4 months in non-diabetic but insulin-resistant HF patients resulted in a significant (*p* < 0.001) improvement in IR indices associated with a significant (*p* < 0.036) reduction of the VE/VCO2 slope [57]. To verify, on a large scale, the effects of treatment with metformin on atherosclerotic cardiovascular outcomes, several large-scale studies have begun, which will end in a few years and will finally tell us with absolute clarity whether this old drug, active against IR/Hyperins, is effective in reducing the risk of death, heart attacks, and/or strokes in patients who have prediabetes and heart or vascular problem [58,59].

Berberine, an isoquinoline alkaloid isolated from the Chinese herb Coptis chinensis and other Berberis plants, has a wide range of pharmacological properties. Berberine has been used for over 2000 years in Chinese and Indian Ayurvedic medicine for its multiple beneficial effects on human health to treat many diseases, such as cancer and digestive, metabolic, cardiovascular, and neurological diseases. Among its various positive effects, the one relating to the reduction of IR/Hyperins has been widely demonstrated in the literature [60,61]. A fair amount of the scientific literature demonstrates that berberine treatment improves endothelial dysfunction, determining beneficial effects on the development and progression of atherosclerosis over time [62,63,64]. A study verified that in mice deficient in apolipoprotein E, berberine regulated 199 proteins involved in numerous biological pathways, including those involved in mitochondrial dysfunction, in β-oxidation I of fatty acids, and in FXR/RXR activation, suggesting that berberine treatment can improve endothelial dysfunction, slowing the evolution of atherosclerosis [62]. Another human study compared 12 healthy subjects treated with berberine for one month with 11 healthy subjects without therapy. Endothelial function was assessed using the flow mediated dilation (FMD) technique on the forearm, also through the administration of sublingual nitroglycerin. Circulating endothelial microparticles (EMPs), which, as is known, produce endothelial dysfunction by increasing oxidative stress, and serum malondialdehyde (MDA) were also measured before and after treatment. The results of the study showed an improvement in FMD with the reduction in MDA and circulating EMPs (CD31+/CD42-MPs) [63].

A recent experimental study in a murine model of HFpEF has shown that berberine alleviates myocardial diastolic dysfunction by modulating the dynamin-related protein1 (Drp1), a GTPase that regulates mitochondrial fission and Ca²^+^ homeostasis [65]. A few years ago, a placebo-controlled study was published that aimed to verify, in patients with HF, the effects of berberine therapy on the heart. The evaluation was carried out at baseline, after 2 months, and after an average follow-up of 24 months. The results of this study demonstrated that berberine therapy in patients with HF resulted in a significant increase in EF and a reduction in the number and complexity of ventricular extrasystoles. Furthermore, after a follow-up period of 2 years, 7 deaths were verified in the berberine arm while 13 deaths were verified in the placebo arm (*p* < 0.02) [66].

In another randomized, double-blind, and placebo-controlled study, the 18-week treatment with a nutraceutical combination containing 500 mg of berberine in a group of patients with metabolic syndrome resulted in a significant (*p* < 0.05) reduction in HOMA-IR and fasting insulin levels, and this was accompanied by a significant reduction in LVM, relative wall thickness and diastolic dysfunction [67]. These results were confirmed in a more recent multicenter randomized, double-blind and placebo-controlled study on a larger number of patients with metabolic syndrome and LVH, in which treatment with 500 mg of berberine per day for 6 months compared to placebo resulted in a significant (*p* < 0.001) reduction of LVM [68]. The absorption of berberine at the intestinal level can be made more constant by the association of berberine with silymarin, which, in addition to enhancing the metabolic effects of berberine, improves its absorption by blocking intestinal P-glycoprotein [69].

## 6. Conclusions

The prevalence of HFpEF is progressively increasing along with IR/Hyperins syndrome, particularly in developed and developing countries. The scientific literature clearly demonstrates that the two conditions are closely related and are progressively increasing hospitalizations, healthcare costs, and deaths. Therefore, we believe that it is mandatory to consider IR/Hyperins as an independent cardiovascular risk factor, in particular, for the development and worsening of HFpEF. Screening for IR/Hyperins in the general population should be advisable in order to promptly identify affected subjects and initiate a preventive treatment strategy.

## Figures and Tables

**Figure 1 biomedicines-12-00806-f001:**
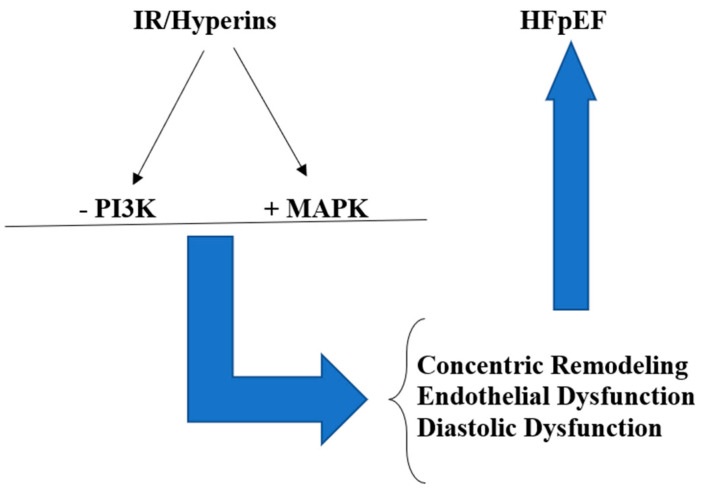
IR/Hyperins and HFpEF are strictly related: Hyperins associated with IR, for the decreased activity (−) of PI3K pathway and increased activity (+) of MAPK pathway, produces concentric remodeling, diastolic dysfunction, and endothelial dysfunction, which, in turn, determine and/or worsen HFpEF.

**Figure 2 biomedicines-12-00806-f002:**
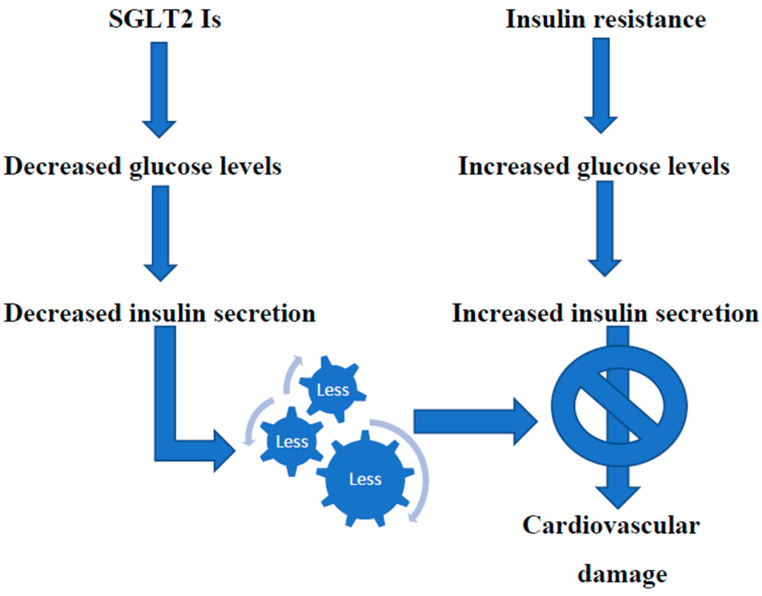
Treatment with SGLT2 Is reduce insulin circulating levels, thus protecting from cardiovascular damage.

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
