# Peer review of "Insulin Resistance/Hyperinsulinemia, Neglected Risk Factor for the Development and Worsening of Heart Failure with Preserved Ejection Fraction"

_biomedicines, 2024, doi:10.3390/biomedicines12040806_

Round 1
Reviewer 1 Report
Comments and Suggestions for Authors
Dear Sir,
This is an interesting manuscript, well designed and well presented.
However, it would be improved if another table and 1-2 figures woyld be added, regarding perhaps the mechanisms of action of SGLT-2 inhibitors or the relationship between IR and HFpEF. in addition, the section of SGLT-2 inhibitors in the treatment of HFpEF could be further elucidated upon with more details regarding the existing studies.
Lines 77-79 should be re-written and they have unnecessary repetitions of words.
Lines 164-165 as well. The sentence from 165-170 is too long to be comprehensive.
Lines 292-293 should be re-written as well.
In addition to my previous comments, the relationship between HF and natriuretic peptide should be further elucidated.
Besides, the significance of SGLT-2 inhibitors, as I have already pointed out, should be further elaborated upon. For example, the cut-off points for administration of SGLT-2 inhibitors among patients with HF and CKD than among patients with DM and CKD should be also stressed.
Finally, references should be added as well to further enhance the quality of the paper.
There are many up to date references that should be discussed and added as well. For example, the 2 following:
1. V I K A S H Jaiswal, S O N G P Ang, M A H A Hameed, J I A E E Chia, K R I T I Kalra, A A Attia, S S Kanakannavar, S Roy, S I D R A Naz, V I C T O R Hugo Alvarez-Aguilera, P R A C H I Sharma, A K A S H Jaiswal, SGLT2 inhibitors among patients with heart failure with preserved ejection fraction: a meta analysis of randomised controlled trials, European Heart Journal, Volume 44, Issue Supplement_1, February 2023, ehac779.041, https://doi.org/10.1093/eurheartj/ehac779.041
2. Clinical Evidence and Proposed Mechanisms of Sodium-Glucose Cotransporter 2 Inhibitors in Heart Failure with Preserved Ejection Fraction: A Class Effect?
Brent Deschaine 1 , Sahil Verma 2 , Hussein Rayatzadeh 2 3 4
PMID: 35846984 PMCID: PMC9272408 DOI: 10.15420/cfr.2022.11
Author Response
To Reviewer 1.
We thank the reviewer for his appreciation and the very useful suggestions, which we have made our own, modifying the manuscript, which we believe has clearly improved.
1R. However, it would be improved if ......
1A. We have added a figure 2 regarding the action of SGLT2 Is on IR/Hyperins.
2R. Lines 77-79 should be rewritten...
2A. We have rewritten and corrected lines 77-79.
3R. Lines 164-165 as well. The sentence from 165-170 is too long....
3A. We have corrected and shortened the period, in order to make it more understandable.
4R. Lines 292-293 should be rewritten...
4A. Lines 292-293 have been rewritten.
5R. In addition to my previous comments, the relationship between HF and natriuretic peptide should be further elucidated.5A. We have added a sentence to better clarify the relationship between HF and natriuretic peptide.6R. Besides, the significance of SGLT2 Is, as I have already pointed out, should be further elaborated upon. For example, the cut-off points for administration....6A. According to the Reviewer's suggestion, we have further elaborated the paragraph on SGLT2 Is, also by adding the evolution of cut-off values ​​of eGFR to be able to start therapy with these drugs.7R. There are many up to date references that should be discussed and added as well. For example, the 2 following...7A. According to the Reviewer's suggestion, we have added and commented other references and, in particular, the 2 important references he has suggested.
Thank you and best regards.

Reviewer 2 Report
Comments and Suggestions for Authors
The authors aimed at reviewing the association of IR/hyperinsulinemia and HFpEF.
The review is not carefuly written, some references used are not adequate and do not reflect the content of the review. There are too many repetitions and missing points that need to be addressed more specifically and more clearly.
1. Abstract
Too long.
lane 32: ..which, coincidentally. Something is missing here.
Chapter 2
lanes 77 - 79: Repetitions
Chapter 4
Molecular mechanisms should be described more precisely.
Lanes 150-162: Refs 27, 28 are not appropriate here; they do not describe the relationship between insulin and adhesion molecules or endothelial dysfunction; please re-writte the section and provide appropriate references.
lane 172: anti-natriuretic effects; please change
into: anti-natriuretic peptides effects
Fig. 1 sheme should be graphically better designed
Lanes 260-281: The part on berberine is poorly written, please try to improve it.
Comments on the Quality of English LanguageEditing required.
Author Response
To the Reviewer 2.
We thank the reviewer for his relevant suggestions. We have revised the manuscript accordingly and think that it is greatly improved.
1R. Abstract, too long.
1A. We agree with the Reviewer, therefore we have shortened and think we have improved the abstract.
2R. Lane 32, which, coincidentally. Something is missing here.
2A. We have corrected the sentence.
3R. Chapter 2. Lanes 77-79: repetitions.
3A. Thank you. Yes, we have corrected the sentence.
4R. Chapter 4. Molecular mechanisms should be described more precisely.
4A. Although it was not the main purpose of this article, we have modified the chapter a little.
5R. Lanes 150-162: Refs 27-28 are not appropriate here....
5A. We thank the Reviewer for noticing the error. We have therefore modified the section and included appropriate references.
6R. Lane 172: anti-natriuretic effects; please change into: anti-natriuretic peptides effects.
6A. We have changed the sentence.7R. Fig. 1 scheme should be graphically better designed.7A. According to the Reviewer's suggestion, we attempted to improve the graph in Figure 1.8R. Lanes 260-281: The part on berberine is poorly written, please try to improve it.8A. We have also attempted to improve the chapter on berberine by adding a comment to a reference to a recently published experimental study on the effects of berberine on HFpEF.
Thank you and best regards.

Round 2
Reviewer 1 Report
Comments and Suggestions for Authors
Dear Sirs,
the manuscript has been substantially improved. However, as I have already asked a second figure or a table stating the pleiotropic properties of SGLT-2 inhibitors should be further added.
Author Response
To the Reviewer 1.
Acording to your kind suggestion we had already attached this figure (figure 2), which shows the possibile effect of reducing insulin levels in patients with insulin resistance by SGLT2 Is. But maybe You didn't see it.
Thank you again and best regards

Reviewer 2 Report
Comments and Suggestions for Authors
Manuscript haas been improved and can be accepted.
Comments on the Quality of English LanguageOnly minors detected.
Author Response
To the Reviewer 2,
Thank you for your decision and best regards